# Affordability of Paediatric Oral Anti-Infective Medicines in a Selected District, Sri Lanka

**Malith Kumarasinghe** [1,*] and **Manuj C. Weerasinghe** [2]

1    Postgraduate Institute of Medicine, University of Colombo, 160, Prof. Nandadasa Kodagoda Mawatha, Colombo 00700, Sri Lanka
2    Department of Community Medicine, Faculty of Medicine, University of Colombo, 25 Kynsey Rd, Colombo 00800, Sri Lanka; manujchri@gmail.com
*    Correspondence: malith.kumarasinghe@yahoo.com

**Abstract:** In this cross-sectional descriptive study conducted in the Ratnapura district, Sri Lanka, we assessed the affordability of oral pediatric anti-infective medicines (OPAIMs). Using a modified WHO/HAI medicinal price methodology, we examined the availability, median price ratios (MPRs), mean percentage difference, and affordability of the standard treatment of the originator brand (OB) and lowest-priced generic (LPG) OPAIMs in 30 private and 2 state-owned pharmacies. The study revealed disparities in availability, with only 50% of private pharmacies offering all 11 medicinal drugs in their generic form. The MPRs of OPAIMs for OB and LPG varied, with three drugs exceeding the financially acceptable MPR of 2 (albendazole, amoxicillin, and erythromycin). The standard treatment with LPGs costs between 0.17 and 0.85 and between 0.06 and 0.28 days' wages for the lowest daily salary of the private sector and unskilled public employees, respectively. We identified erythromycin and albendazole as having less than 50% availability in their generic form in private pharmacies. To address these findings, we recommend frequent pricing revisions based on exchange rates and associated costs, coupled with the establishment of a transparent scientific criterion to subsidize essential medicines deemed "unaffordable." Failure to implement such measures amidst economic crises may adversely impact financial access to essential medications.

**Keywords:** paediatric; anti-infective; medicines; affordability; Sri Lanka





## 1. Introduction

Affordability, often referred to as financial access in the literature, presents a complex challenge in the space of healthcare. The literature offers various methodologies and techniques to evaluate the financial accessibility of medicines, given that affordability is intricately tied to numerous influencing factors [1–3]. The ability to afford a specific medication can vary significantly based on an individual's socioeconomic status, rendering a medicine affordable to one person but financially difficult to another [4]. Determining financial access to medicines demands a nuanced and concentrated effort due to these multifaceted dynamics. Various techniques are employed, including retrospective assessments of individuals who have experienced impoverishment or incurred catastrophic expenses as a result of healthcare costs [2]. Organizations such as the World Health Organization (WHO) and Health Action International (HAI) have contributed to this endeavor through their essential medicinal drug pricing initiatives. These methodologies collectively contribute to a comprehensive understanding of the intricate landscape of financial access to medicine [2,5]. However, it is important to understand that each of these techniques has limitations. The WHO/HAI medicinal price methodology used in this study includes estimation by an index termed the median price ratio (MPR) for the "originator brand" (OB) and "lowest price generic" (LPG) equivalents and by the average salary of the lowest-paid unskilled public worker. The MPR is the ratio between the median local price and the international reference unit price [6,7].

A national survey on the price and affordability of key essential medicines for children in Sri Lanka revealed poor availability in public hospitals and poor affordability in private institutions and pharmacies. This situation could cause poor control of specific diseases in children, which is unacceptable. Thus, affordability remains a key issue in Sri Lanka [6]. Furthermore, the survey revealed that due to disparities in income in communities and improper treatment, the true situation may not be fully understood. Therefore, further studies are needed to explore the forces at work that determine access to medicines in a given community. They may not be evident in a national study where specific issues in a specific community may be obscured by the overall picture [6].

The Ratnapura District in Sri Lanka is unique geographically, culturally, and economically. According to the Demographic and Health Survey or DHS (2016) [8], Ratnapura District reported the highest under-five child mortality rate (outside the Northern and Eastern Provinces) in Sri Lanka. It has a good representation of the urban (9.1 percent), rural (81.7 percent), and estate (9.2 percent) sectors [8]. It has hilly areas with difficult access, as well as low-altitude areas with easy access. Therefore, Ratnapura was selected as the study district in Sri Lanka.

The affordability of anti-infectives, especially in paediatric dosage forms, is a major issue in Sri Lanka. Therefore, the objective of this study was to describe the affordability of paediatric oral anti-infective medicines in the Ratnapura district, Sri Lanka, immediately prior to the economic recession.

## 2. Results

Among the 11 anti-infective medicinal drugs surveyed, OBs were available for 5 of them in at least 1 of the 32 surveyed private and Osu-Sala pharmacies (Table 1). The antibiotics used were amoxicillin, amoxicillin and clavulanic acid, cephalexin, erythromycin, and albendazole. None of the surveyed pharmacies reported at least a single generic entity for nitrofurantoin and fluconazole among the 32 surveyed private and Osu-Sala pharmacies. In summary, for nitrofurantoin and fluconazole, both OB and generic reported zero overall availability. Among Osu-Sala, only the OBs of amoxicillin and amoxicillin and clavulanic acid were available. In 50% of the private pharmacies, only 7 out of the 11 medicinal drugs were available in their generic form. Erythromycin and albendazole were less than 50% available for generics in private retail pharmacies. Among the Osu-Sala pharmacies, no generics were available for amoxicillin and clavulanic acid, metronidazole, or albendazole.

Table 2 shows the MPRs for OB and LPG for the private sector and Osu-Sala pharmacies. The MPRs for OB ranged from 17.63 to 0.28. The highest MPR for OB was reported for albendazole, whereas the lowest MPR was reported for cephalexin. The cutoff value for the MPR according to the financially acceptable level was two. Among the five medicinal drugs for which the MPRs for OB were calculated, three were reported with MPRs of more than two, namely, albendazole, amoxicillin and erythromycin. The MPRs for LPG ranged from 8.85 to 0.22. The highest MPR for LPG was reported for pyrantel, whereas the lowest MPR was reported for cephalexin. Among the nine medicinal drugs for which the MPRs for LPG were calculated, three were reported with an MPR of more than 2, namely, albendazole, metronidazole, and Pyrantel. Six medicinal drugs out of nine (albendazole, acyclovir, amoxicillin, erythromycin, metronidazole, and pyrantel) were reported with MPRs greater than one (the price of LPG was greater than that of the IRP). The mean percentage difference (OB vs. LPG) ranged from 120.59 to 6.38. The greatest difference in the mean percentage was observed for albendazole, while the lowest difference was observed for amoxicillin and clavulanic acid. The mean percentage difference of three of the calculated medicinal drugs was greater than 100%. Hence, for these three medicinal drugs, the OB cost is more than twice the cost of LPG. Among the Osu-Sala pharmacies, the MPRs for OB were reported only for amoxicillin and amoxicillin and clavulanic acid. An MPR of more than two was reported for amoxicillin. The MPRs for LPG ranged from 8.72 to 0.33. The highest MPR for LPG was reported for Pyrantel, whereas the lowest MPR was reported for cephalexin. Among the six medicinal drugs for which the MPRs for LPG

were calculated, three were reported with MPRs of more than two, namely, amoxicillin, erythromycin, and pyrantel (50% of available medicinal drugs).

**Table 1.** Availability of selected essential paediatric oral anti-infective medicines in the Ratnapura District.

| Medicine | Originator Brand (OB) | | | Generic | | |
|---|---|---|---|---|---|---|
| | Private (*N* = 30) | Osu-Sala (*N* = 2) | Total | Private (*N* = 30) | Osu-Sala (*N* = 2) | Total |
| Amoxicillin suspension 125 mg/5 mL (100 mL) | 21 | 2 | 23 | 20 | 1 | 21 |
| Amoxicillin and Clavulanic acid suspension 125 mg + 31.25 mg/5 mL (100 mL) | 27 | 2 | 29 | 17 | 0 | 17 |
| Cephalexin suspension 125 mg/5 mL (100 mL) | 2 | 0 | 2 | 28 | 2 | 30 |
| Cloxacillin syrup 125 mg/5 mL (100 mL) | 0 | 0 | 0 | 22 | 2 | 24 |
| Erythromycin syrup 125 mg/5 mL (100 mL) | 24 | 0 | 24 | 1 | 2 | 3 |
| Metronidazole suspension 200 mg/5 mL (100 mL) | 0 | 0 | 0 | 16 | 0 | 16 |
| Nitrofurantoin 25 mg/5 mL (100 mL) | 0 | 0 | 0 | 0 | 0 | 0 |
| Acyclovir 200 mg (tablet) | 0 | 0 | 0 | 22 | 2 | 24 |
| Albendazole 200 mg (tablet) | 17 | 0 | 17 | 8 | 0 | 8 |
| Pyrantel suspension 50 mg/mL (10 mL) | 0 | 0 | 0 | 28 | 1 | 29 |
| Fluconazole liquid 50 mg/5 mL (35 mL) | 0 | 0 | 0 | 0 | 0 | 0 |

**Table 2.** Median price ratios of surveyed medicines and mean percentage differences between the originator brand and the lowest-priced generic medicines in the Ratnapura District.

| | Private Sector Pharmacies | | | Other Sector/Osu-Sala Outlets | | |
|---|---|---|---|---|---|---|
| | Originator Brand MPR | Lowest Priced Generic MPR | Mean Percent Difference (OB vs. LPG) | Originator Brand MPR | Lowest Priced Generic MPR | Mean Percent Difference (OB vs. LPG) |
| Albendazole 200 mg tablet | 17.63 | 7.99 | 120.59 | NA | NA | - |
| Acyclovir 200 mg tablet | NA | 1.16 | - | NA | 0.94 | - |
| Amoxicillin 125 mg/5 mL | 2.93 | 1.43 | 104.55 | 2.93 | 2.71 | 8.12 |
| Amoxicillin and Clavulanic acid 125 mg + 31.25 mg/5 mL | 0.85 | 0.80 | 6.38 | 0.85 | NA | - |
| Cephalexin 125 mg/5 mL | 0.28 | 0.22 | 27.89 | NA | 0.33 | - |
| Cloxacillin 125 mg//5 mL | NA | 0.95 | - | NA | 0.95 | - |
| Erythromycin 125 mg/5 mL | 2.73 | 1.34 | 104.55 | NA | 2.62 | - |
| Fluconazole 50 mg/5 mL | NA | NA | - | NA | NA | - |
| Metronidazole 200 mg/5 mL | NA | 3.58 | - | NA | NA | - |
| Nitrofurantoin 25 mg/5 mL | NA | NA | - | NA | NA | - |
| Pyrantel 50 mg/mL * | NA | 8.85 | - | NA | 8.72 | - |

\* For pyrantel, the international reference price (IRP) was not available. Therefore, India was selected as the reference country. India was selected because it is the geographically closest country to Sri Lanka and the only Asian country among the top 10 pharmaceutical markets by volume in the world (ranked third) [9]. All Pyrantel brand prices available online in India were obtained [10–12]. The median price was calculated from these available prices. This median price was adopted as the IRP for the calculation of MPRs for Pyrantel. Mean percentage difference = MPR of OB—MPR of LPG/MPR of LPG × 100. NA—Not available.

Table 3 shows the cost of standard treatment with anti-infective medicinal drugs in the private sector and Osu-Sala pharmacies according to the number of days of income of a Sri Lankan worker. The daily salary for the private sector according to the Sri Lankan Law was 400 LKR in 2016 [13]. The lowest daily salary of an unskilled public employee is LKR 1213.67 [14]. Standard treatment with OB or the highest-priced brand of the selected medicinal drugs costs 0.38 to 3.89 and 0.12 to 1.28 days' wages of the lowest daily salary of the private sector and unskilled public employees, respectively. The highest cost for standard treatment with OB or the highest-priced brand was reported for acyclovir, and the lowest was reported for albendazole. Except for Acyclovir, only erythromycin costs more than 1 day's wage in the private sector for standard treatment with OB or the highest-priced brand. Standard treatment with the LPG of the selected medicinal drugs costs between 0.17 and 0.85 and between 0.06 and 0.28 days' wages for the lowest daily salary of the private sector and unskilled public employees, respectively. The highest cost for standard treatment with LPG was reported for amoxicillin and clavulanic acid, and the lowest was reported for albendazole. Standard treatment with all available LPGs of the selected medicinal drugs costs less than a single day's wage of the lowest daily salary of the private sector and unskilled public employees [13,14].

**Table 3.** Cost of standard treatment with oral anti-infective medicines (to purchase the medicine only) as the number of days of income for a Sri Lankan worker *.

| | Private Sector Pharmacies | | | | | | Other Sector/Osu-Sala Outlets | |
|---|---|---|---|---|---|---|---|---|
| | Lowest Daily Salary for Private Sector by Sri Lankan Law | | Lowest Daily Salary of Unskilled Public Employee | | Lowest Daily Salary for Private Sector by Sri Lankan Law | | Lowest Daily Salary of Unskilled Public Employee | |
| | Originator Brand | Lowest Priced Generic | Originator Brand | Lowest Priced Generic | Originator Brand | Lowest Priced Generic | Originator Brand | Lowest Priced Generic |
| Albendazole 200 mg tablet | 0.38 | 0.17 | 0.12 | 0.06 | NA | NA | NA | NA |
| Acyclovir 200 mg tablet | 3.89 | 0.68 | 1.28 | 0.22 | NA | 0.55 | NA | 0.18 |
| Amoxicillin 125 mg/5 mL | 0.71 | 0.35 | 0.23 | 0.11 | 0.71 | 0.66 | 0.23 | 0.11 |
| Amoxicillin and Clavulanic acid 125 mg + 31.25 mg/5 mL | 0.91 | 0.85 | 0.30 | 0.28 | 0.91 | NA | 0.30 | NA |
| Cephalexin 125 mg/5 mL | 0.71 | 0.55 | 0.23 | 0.18 | NA | 0.83 | NA | 0.28 |
| Cloxacillin 125 mg/5 mL | NA | 0.65 | NA | 0.21 | NA | 0.65 | NA | 0.21 |
| Erythromycin 125 mg/5 mL | 1.58 | 0.77 | 0.52 | 0.25 | NA | 1.51 | NA | 0.50 |
| Fluconazole 50 mg/5 mL | NA | NA | NA | NA | NA | NA | NA | NA |
| Metronidazole 200 mg/5 mL | NA | 0.51 | NA | 0.17 | NA | NA | NA | NA |
| Nitrofurantoin 25 mg/5 mL | NA | NA | NA | NA | NA | NA | NA | NA |
| Pyrantel 50 mg/mL | NA | 0.40 | NA | 0.13 | NA | 0.39 | NA | 0.13 |

* Anti-infective regimes for affordability are given in Supplementary File S3. NA—Not available.

Among Osu-Sala pharmacies, standard treatment with LPG from the selected medicinal drugs costs between 0.39 and 1.51 and between 0.11 and 0.5 days' wages for the lowest daily salary of the private sector and unskilled public employees, respectively. The highest cost for standard treatment with LPG was reported for erythromycin, and the lowest was reported for pyrantel. In the standard treatment with all available LPGs for the selected medicinal drugs, the lowest daily salary of the private sector and unskilled public employees was less than the cost of a single day, except for erythromycin, which required the lowest daily salary of the private sector (1.51 days' wages).

Figure 1 illustrates the affordability vs. availability of LPG medicinal drugs in the private sector. Except for erythromycin and Pyrantel, a pattern is evident where high availability was observed for increased affordability (low MPRs), and low availability was observed for decreased affordability (high MPRs).

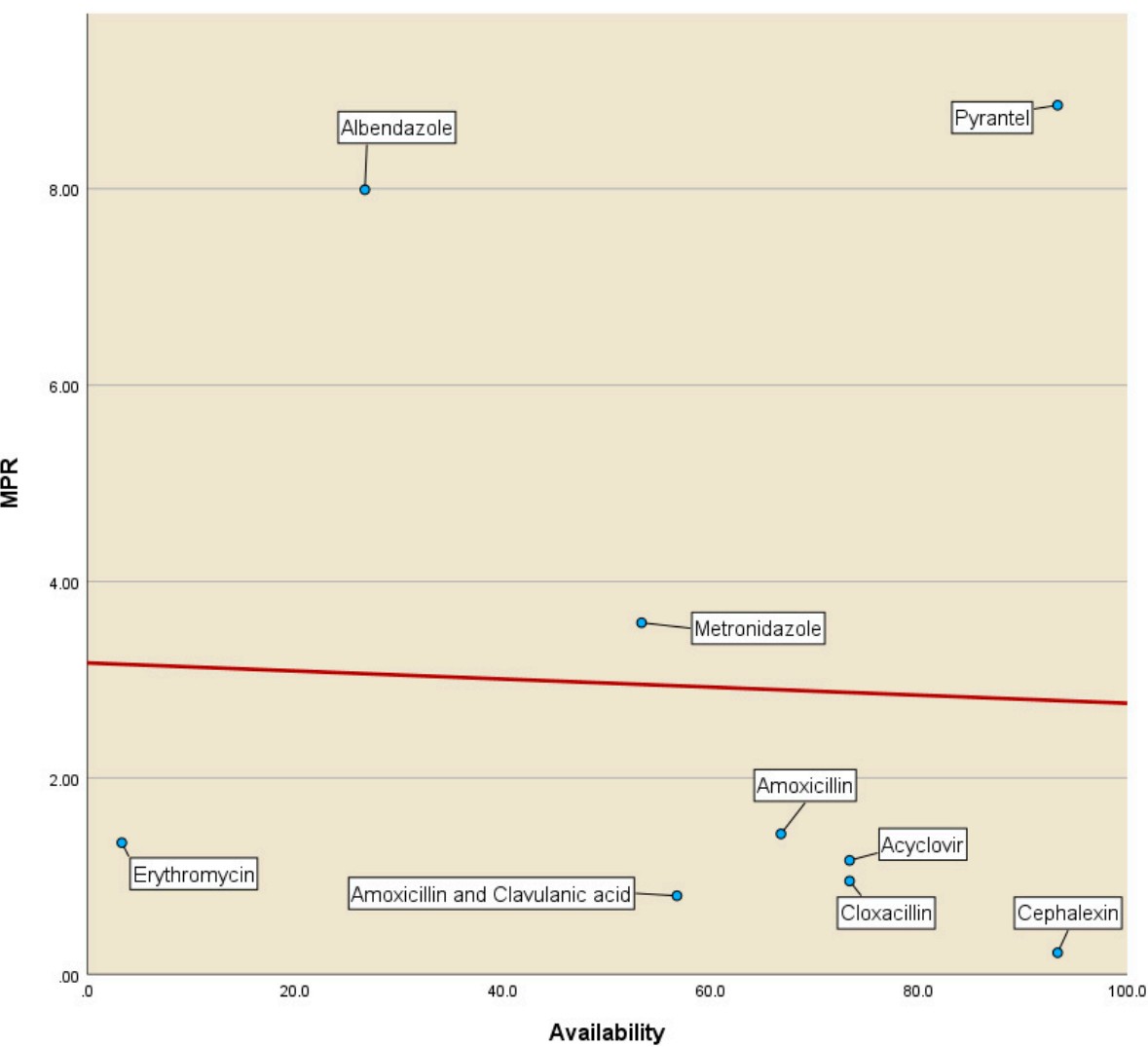

**Figure 1.** Affordability vs. availability of LPGs of anti-infective medicines in the private sector (MPR—median price ratio).

### 3. Discussion

We used the private and other sector retail price components of the WHO/HAI medicinal price methodology to assess the affordability of oral paediatric anti-infective medicinal drugs in the Ratnapura District of Sri Lanka. We chose the WHO/HAI medicinal price methodology because it provides a uniform and methodologically sound system for monitoring the affordability and availability of medicinal drugs in a given country or region. It also provides the opportunity for comparison across countries. However, this study only included two "Osu-Sala" pharmacies as the "other" sector for the assessment of financial access to paediatric anti-infective medicines. The selection of the two available "Osu-Sala" pharmacies could have limited the validity of the findings of the medicinal prices and the affordability of paediatric oral anti-infective medicines to primary caregivers in the "other" sector. However, the number of "Osu-Sala" pharmacies surveyed might not have influenced the findings on oral paediatric medicinal prices and affordability assessment

to a greater extent in the "other" sector category because of the central procurement and central retail pricing of medicines practiced at the "Osu-Sala" franchise [15].

Four key essential medicines demonstrated poor availability of less than 50% for generics in the private sector (Table 1). This could potentially impact the access to affordable and essential medicines for children for common infections and cause significant harm to the paediatric population in the district. Further, the unavailability of nitrofurantoin in liquid form (either generic or OB), reported in both private and Osu-Sala pharmacies, could pose a significant risk to the health of children in the district (Table 1). Nitrofurantoin is one of the first-line treatments for uncomplicated lower urinary tract infections (UTIs) in children and for prophylaxis from UTIs [16]. Similarly, the availability of metronidazole in liquid form in the district is less than 50% (Table 1). Metronidazole is being used as one of the first-line treatments for mild-to-moderate Clostridium difficile-associated disease in children, which is a common cause of gastroenteritis in children [17]. Both metronidazole and nitrofurantoin are listed as essential medicines by the WHO, underlining the importance of universal availability for the treatment of children, and nonavailability is a public health issue that requires urgent intervention [18].

We observed that in four out of nine key essential anti-infective paediatric medicines (excluding nitrofurantoin and fluconazole, which demonstrated zero availability for both OB and generics in private pharmacies), the availability of OB was higher than the generics. Conversely, generic availability exceeded OB availability in only five medicines (Table 1). This prompts the question of whether the scarcity of low-cost generics might force patients to purchase relatively expensive OB drugs [19]. Further investigation is needed to assess this situation in other districts of Sri Lanka, as it could potentially hinder access to essential paediatric medications for the most vulnerable communities in the country. The U.S. Food and Drug Administration has stated that 63% of the total drug shortages between 2013 and 2017 in the USA were due to generic medicines, highlighting that the issue is not restricted to low-income countries [19]. Therefore, similar studies on affordability conducted in different geographical regions during the post-COVID-19 era could provide valuable insight into the current situation in the respective countries.

Few studies conducted in Asia have reported the availability of OB and generic oral paediatric anti-infective medicinal drugs in the private sector. Two studies conducted in two different provinces in China reported zero availability of the OB of amoxicillin and clavulanic acid suspension [20,21]. For the generic form, a study conducted by Wang et al. (2014) reported zero availability, whereas a 2017 study reported 3.3% availability [20]. Our study reported 96.4% and 60.7% availability with respect to OB and generic forms, respectively (Table 1). In addition, for 200 mg of acyclovir tablets, Sun et al. (2018) reported (in a study conducted in 2017) zero and 6.7% availability with respect to OB and generic forms, respectively, whereas our study reported zero availability for OB [21]. However, the better availability of the generic form was noted in our study for acyclovir (78.6%). Wang et al. (2014) revealed the better availability of the OB of albendazole compared to our study (70% vs. 60.7%). However, for the generic product, our study reported better availability than that reported by Wang et al. (2014) (28.6 vs. 6.7 percent) [20]. The aforementioned studies have documented the availability of the mentioned medicines, dating back at least five years. Recent evidence indicates a notable increase in the accessibility of these medicines in China, particularly in the years after the implementation of the new medicine policy in 2015. The discernible positive effects of this policy shift became evident from 2018 onward [22].

Comparing the MPRs of studies conducted in the African region with our study findings, a key observation was the comparatively lower price (MPR) in Sri Lanka for the LPG of oral paediatric medicinal drugs, namely, amoxicillin, amoxicillin, and clavulanic acid, compared to those in Tanzania (amoxicillin: 1.73 vs. 1.43) and Ethiopia (amoxicillin: 2.07 vs. 1.43; amoxicillin and clavulanic acid: 1.52 vs. 0.80) [23,24]. However, compared to Kenya, the MPR of the LPG of amoxicillin is greater in Sri Lanka (1.1 vs. 1.43) [25]. The lower medicinal prices in Kenya than in Sri Lanka and other African countries may be due to the local production of pharmaceuticals. Kenya is one of the leading pharmaceutical

manufacturing countries on the African continent (others are South Africa and Nigeria) [26]. Despite being a leading pharmaceutical manufacturing country, Kenya did not report the lowest MPR for the LPG of metronidazole syrup (Tanzania reported the lowest MPR: 1.7689 vs. 1.9) [23,25]. Compared to all the studies conducted in Africa, our study revealed the highest MPR for the LPG of metronidazole syrup, which was higher than those of both Kenya and Tanzania (3.58 vs. 1.9 and 1.7689) [23,25]. Similarly, for erythromycin, the MPR of LPG was greater in Sri Lanka than in Tanzania (1.34 vs. 1.0417) [23]. A notable observation from our study was the substantially lower MPR for OB than for Africa. In Kenya, a greater than 500 percent mean percentage difference was observed for the two selected essential paediatric oral anti-infective medicinal drugs, whereas none of the surveyed medicinal drugs in our study exhibited a mean percentage difference of 500 percent or more [25].

In India, few studies have examined the affordability of oral paediatric anti-infective medicinal drugs in recent years. Compared to the two studies that were confined to a state in India, namely, Chhattisgarh (2009) and Odisha (2015), Sri Lanka showed lower MPRs for the LPGs of oral paediatric anti-infective medicinal drugs, namely, amoxicillin (Chhattisgarh and Odisha; 4.1 and 3.83 vs. 1.43) and amoxicillin and clavulanic acid (Chhattisgarh; 1.48 vs. 0.85) [27,28]. Interestingly, the MPR for amoxicillin was lower than half of that of Odisha or Chhattisgarh [27,28]. Similarly to the comparison with Africa, our MPR for the LPG of metronidazole syrup was greater than that of India (Chhattisgarh; 1.27 vs. 3.58) [28]. The MPR of metronidazole in Sri Lanka displayed a close to 300% increase compared to the MPR in Chhattisgarh [28]. However, it should be noted that each of the three compared studies was carried out at 6-year intervals. Therefore, the time factor could also have played a part in the observed difference. In addition, both Chhattisgarh and Odisha are poor states (26th and 28th out of 33 states in per capita income, respectively). Therefore, the generalisability of this comparison to India as a whole may not reveal the true picture [29].

The main study conducted in Sri Lanka, prior to the present study on the affordability of oral paediatric anti-infective medicinal drugs, was the "National Survey of Price and Affordability of Key Essential Medicines for Children in Sri Lanka", which was conducted in 2009 and repeated in 2018/2019 [6,30]. Table 4 compares the MPRs of OB and LPG for selected essential anti-infective medicinal drugs. The key observation in Table 4 is the reduction in the MPRs of the OBs of amoxicillin and amoxicillin and clavulanic acid from 2009 to 2021. Only erythromycin increased the MPR of the OB. However, all three medications (amoxicillin, amoxicillin and clavulanic acid, and erythromycin) reduced the MPR for the OB from 2018/2019 to 2021. In addition, the MPRs of the LPG containing amoxicillin and amoxicillin and clavulanic acid increased, whereas the MPRs of the LPG containing dicloxacillin and erythromycin decreased from 2009 to 2021. However, all three medications (amoxicillin, amoxicillin and clavulanic acid, and erythromycin) reduced the MPR for LPG from 2018/2019 to 2021 [6,30]. One possible reason for this may be the recently introduced maximum retail price control. In 2016, the National Medicines Regulatory Authority introduced maximum retail prices for 48 selected medicinal product formulations. This was last revised in the year 2019, and the regulatory price control was expanded to 60 selected medicinal product formulations. These formulations include both adult and paediatric preparations. Table 5 summarises the regulated oral paediatric anti-infective preparations and the preparations that were not regulated but included in our study [31]. There were three oral paediatric anti-infective medicinal drugs that were included in our study, and their price was regulated (for the same dosage form). They are acyclovir, amoxicillin clavulanic acid, and cephalexin. The common characteristic observed in these three medicinal drugs was that the MPRs of available OB and LPG were below the acceptable price level of 2. The MPRs of the OB of all price-controlled medicinal drugs (except for acyclovir, for which the MPR of the OB was not available) were less than one. In other words, the MPRs of the OBs of these medicinal drugs were below the International Reference Price. Therefore, the prices of these medicinal drugs are acceptable. Among

the five studied oral paediatric anti-infective medicinal drugs for which MPRs for OB were available, three preparations were not price controlled. Compared to price-controlled OB preparations, these three OB preparations, namely, amoxicillin, erythromycin, and albendazole, had higher MPRs. The MPRs of all three medicinal drugs are above 2, which is above the cutoff for the acceptable price. Thus, their prices are "unacceptable" or expensive (Table 5).

**Table 4.** Comparison of the MPRs of OB and LPG from selected essential oral paediatric anti-infective medicines in Sri Lanka in 2009 and 2018/2019 with those of the Ratnapura District 2021.

| | 2009 [6] | | | 2018/2019 [30] | | | 2021 | | |
|---|---|---|---|---|---|---|---|---|---|
| | Originator Brand MPR | Lowest Priced Generic MPR | Mean Percent Difference (OB vs. LPG) | Originator Brand MPR | Lowest Priced Generic MPR | Mean Percent Difference (OB vs. LPG) | Originator Brand MPR | Lowest Priced Generic MPR | Mean Percentage Difference (OB vs. LPG) |
| Amoxicillin 125 mg/5 mL | 3.37 | 1.37 | 146.38 | 3.89 | 2.59 | NA | 2.93 | 1.43 | 104.55 |
| Amoxicillin and Clavulanic acid 125 mg + 31.25 mg/5 mL | 1.08 | 0.77 | 40 | 0.94 | 0.93 | NA | 0.85 | 0.80 | 6.38 |
| Cloxacillin 125 mg/5 mL | NA | 1.31 | NA | NA | 0.79 | NA | NA | 0.95 | NA |
| Erythromycin 125 mg/5 mL | 2.56 | 2.09 | 22.22 | 3.27 | 1.45 | NA | 2.73 | 1.34 | 104.55 |

**Table 5.** MPRs of OB and LPGs of essential paediatric oral anti-infective medicinal preparations regulated in Sri Lanka vs. those selected for the study *.

| | Included in the Study | Price Controlled in Same Form and Strength | Price Controlled in Different Form or Strength | Originator Brand Median Price Ratio | Lowest Price Generic Median Price Ratio |
|---|---|---|---|---|---|
| Acyclovir 200 mg | Yes | Yes | Not Applicable | Not Available | 1.16 |
| Amoxicillin 125 mg/5 mL | Yes | No | No | 2.93 | 1.43 |
| Amoxicillin and Clavulanic Acid 125 mg + 31.25 mg/5 mL | Yes | Yes | Not Applicable | 0.85 | 0.80 |
| Albendazole 200 mg | Yes | No | Yes—400 mg | 17.63 | 7.99 |
| Cephalexin 125 mg/5 mL | Yes | Yes | Not Applicable | 0.28 | 0.22 |
| Cloxacillin 125 mg/5 mL | Yes | No | No | Not Available | 0.95 |
| Erythromycin 125 mg/5 mL | Yes | No | No | 2.73 | 1.34 |
| Metronidazole 200 mg/5 mL | Yes | No | No | Not Available | 3.58 |
| Pyrantel 50 mg/mL | Yes | No | No | Not Available | 8.85 |
| Fluconazole 50 mg/5 mL | Yes | No | Yes—50 mg tablet | Not Available | Not Available |
| Nitrofurantoin 25 mg/5 mL | Yes | No | No | Not Available | Not Available |
| Azithromycin 200 mg/5 mL | No | Yes | Not Applicable | Not Applicable | Not Applicable |
| Clarithromycin 125 mg/5 mL | No | Yes | Not Applicable | Not Applicable | Not Applicable |
| Cefixime 100 mg/5 mL | No | Yes | Not Applicable | Not Applicable | Not Applicable |

* National Medicines Regulatory Authority, 2021.

The case of amoxicillin vs. amoxicillin and clavulanic acid warrants further discussion. Both of these medicinal drugs belong to the same penicillin medicinal drug class, as amoxicillin and clavulanic acid are newer members of the penicillin medicinal drug class than amoxicillin. The maximum retail price of amoxicillin is not controlled, whereas the combination of amoxicillin and clavulanic acid is price-controlled. This could have resulted in increased MPRs of both the OB and LPG of "older" amoxicillin compared to "newer" amoxicillin and clavulanic acid oral paediatric preparations (Table 5) [32].

The abovementioned facts of the unacceptable MPRs of OBs that were not price-controlled point toward a positive impact of the price control of surveyed essential medicinal drugs in the country. However, there could also be negative consequences. For example, the significant depreciation of the Sri Lankan Rupee against the US dollar and subsequent

denial by the governing body to permit an increase in the upper threshold of prices for price-controlled essential medicinal drugs were identified as possible causes by the All Ceylon Private Pharmacy Owners' Association for the reduced availability of some of the essential medicinal drugs in the private sector in Sri Lanka during the month of June 2021 [33,34]. Therefore, alterations to the present price control method may enhance the positive impact and reduce the negative impact of price control for essential medicinal drugs in Sri Lanka. First, price-controlled drugs can be revised annually. Although countries such as the UK review prices once every 5 years, frequent fluctuations in SL rupees against the US dollar warrant the abovementioned price adjustment to reduce the disruption of supply chains and to maintain the status quo [34–36]. Second, it is important to develop a criterion to subsidise essential medicinal drugs. This could be carried out in two ways. First, the retail prices of individual essential medicinal drugs could be controlled. For example, if the MPR exceeds 2 or 2.5, a scheme to make medicinal drugs affordable for the population could be implemented. Alternatively, the maximum price for individual medicinal drugs could be set separately for branded and generic medicines, as practised in the UK [35,36]. The second approach involves subsidizing the cost of treatment or prescription when it surpasses a specified threshold rather than establishing an upper price limit for a particular medicinal drug. This method is employed in the United States [35]. However, implementing such a system in a country such as Sri Lanka raises practical concerns. This is primarily due to the involvement of multiple stakeholders, including insurance companies, pharmaceutical manufacturing and importing entities, retail medicinal drug stores, hospitals, and local and central public institutions. Achieving timely, clear coordination and agreement among these stakeholders becomes a formidable challenge. Moreover, Sri Lanka differs from other countries in its provision of free health services at the point of delivery in public health institutions and its limited health insurance coverage among the population [37]. As a result, adopting a subsidy-based system in Sri Lanka necessitates careful consideration of its feasibility within the existing healthcare infrastructure and financial landscape.

A significant increase in public salaries since 2019 has been observed in Sri Lanka [14]. Consequently, the affordability of standard treatment with amoxicillin for a child (aged 3 to 5 years) in relation to the lowest daily wage of an unskilled public employee has improved compared to that in the years preceding 2019. The salary adjustment aimed to align with the inflation rate in Sri Lanka, addressing the prior period of a fixed public pay scale. The cost of standard treatment with both OB and LPG of amoxicillin has decreased by half since 2018/2019 in Sri Lanka (compared with the national survey in 2018/2019) [30] (Table 6). This was observed despite amoxicillin not being a price-controlled medicine in Sri Lanka. Although public workers constitute a significant proportion of the workforce in Sri Lanka (13.8%), these findings might not reveal the true picture of the affordability of standard treatment for paediatric acute infections in Sri Lanka. The majority of the Sri Lankan workforce is employed in the private (15.4%) and informal (66.7%) sectors [38,39]. At present, a better comparison of affordability could be carried out with the lowest daily salary of the private sector stipulated by Sri Lankan labour law rather than the lowest daily salary of an unskilled public employee [13]. The standard treatment of amoxicillin was still affordable for workers with the lowest salary in the private sector. A national survey in 2009 and 2018/2019 did not analyse the affordability of treatment with oral paediatric anti-infective medicinal drugs against the lowest daily salary of the private sector stipulated by law [6,30]. Therefore, any direct comparison of the affordability of standard treatment with oral paediatric anti-infective medicinal drugs in the national survey of 2009 and 2018/2019 was limited to the affordability against the lowest daily salary of an unskilled public employee. Nevertheless, as Table 6 suggests, the affordability of standard treatment with amoxicillin could be lower at present than it was 5 to 10 years ago in Sri Lanka [6] (Table 6).

**Table 6.** Comparison of the affordability of standard treatment with amoxicillin (125 mg/5 mL) for acute respiratory infection of a child in Sri Lanka in 2009 and 2021.

| | | Cost of Treatment as Number of Days of Earning | |
|---|---|---|---|
| | | Treatment with Originator Brand | Treatment with Lowest Priced Generic |
| 2009—National Survey [6] | Lowest daily salary of unskilled public employee | 0.53 | 0.22 |
| 2018/19—National Survey [30] | Lowest daily salary of unskilled public employee | 0.54 | 0.36 |
| 2021—Present study | Lowest salary for the private sector by Sri Lankan Law | 0.71 | 0.35 |
| | Lowest daily salary of unskilled public employee | 0.23 | 0.11 |

This study, which was undertaken in 2021 during the later phases of the COVID-19 pandemic and preceded the prevailing economic crisis in Sri Lanka, is significant against the backdrop of the current Sri Lankan socioeconomic scenario. The ongoing economic crisis, exacerbated by the substantial devaluation of the rupee against the US dollar and the rising cost of living in Sri Lanka, has likely intensified challenges related to the affordability of medicines for the following reasons. During the Sri Lankan economic crisis, the national currency was significantly devalued. This can lead to an increase in the cost of imported medicines, as the reduced value of local currency makes foreign purchases more expensive [34]. Economic crises lead to high inflation rates. Inflation can impact the overall cost of living, including healthcare expenses. Medicine prices rise in response to increased production and distribution costs [40]. Economic downturns disrupt supply chains, affecting the availability of raw materials and the transportation of goods [41]. This disruption may lead to shortages and increased prices for certain medicines. Furthermore, economic challenges strain public health systems, leading to resource shortages and difficulties in providing essential healthcare services. This can indirectly impact the accessibility and affordability of medicines for the population [42]. Consequently, the timing of this study, which was conducted just before the onset of the economic crisis, provides a valuable baseline for evaluating and contrasting the current situation in the country. The use of the WHO/HAI medicinal price methodology increases the comparability of this study with similar studies conducted globally. However, this was a cross-sectional study and therefore could not assess the trends in price affordability over time. Furthermore, this study does not reflect the current medical prices and affordability due to the recent hyperinflation experienced in Sri Lanka, which may have resulted in comparatively higher medicinal prices at present [43]. Further, we limited ourselves to 11 key essential paediatric anti-infective medicines and excluded the WHO's recommended essential medicines, such as azithromycin, clarithromycin, and sulfamethoxazole + trimethoprim [18]. This decision was based on the "National List of Essential Medicines Sri Lanka (4th edition)" [44] and the opinion of the expert committee on the selection of anti-infective medicines for the study.

**4. Materials and Methods**

A cross-sectional descriptive study using a modified WHO/HAI medicinal price methodology was carried out among registered private pharmacies to assess the affordability of oral paediatric anti-infective medicinal drugs in the Ratnapura District in Sri Lanka. We excluded private retail pharmacies that did not have a valid up-to-date registration by the National Medicines Regulatory Authority during the study period, as the WHO/HAI medicinal price methodology specifically states that they include only registered pharmacies [7].

The second edition of the WHO/HAI methodology was introduced in 2008 under the title "Measuring medicine prices, availability, affordability and price components". Affordability was expressed as the median price ratio for each medicinal drug. This was calculated as the ratio between the median price of a specific medicinal drug in the surveyed area (local) and the international reference price of the same medicine. This could be disaggregated for LPG and OB (previously known as innovator brands). The WHO recommends the use of an international reference price index in the publication

produced by Management Science for Health (MSH) [45]. The affordability of medicines was further assessed by calculating the number of days' wages required to complete a course of treatment (for acute illnesses, 5 to 7 days; for chronic illnesses, 1 month). The "day's wage" for calculations was taken as the daily wage of the lowest-paid unskilled public worker [7,45].

In the WHO/HAI system, the prices of four main groups are usually included: public institution procurement prices, public institution patient prices, private institution patient prices, and other sector patient prices (in Sri Lanka, the State Pharmaceutical Cooperation (SPC), which is a government corporate entity that maintains a national chain of retail pharmacies) [7,46]. The WHO/HAI medicinal price methodology provides the opportunity for country-by-country comparisons. In addition, WHO/HAI experts provided technical assistance for many countries to conduct their national/provincial surveys using WHO/HAI price methodology, thus improving the quality and standard of the results. In addition, to improve the transparency of prices, on the HAI/WHO webpage, a database of results of all surveys conducted under this methodology was established [47]. The patient prices of public institutions and public institutions' procurement prices were not included in this study, as public institutions provide medications free of charge at the point of delivery to the people of Sri Lanka [7]. Only the patient price components of private institutions and "other" sectors of the second edition of the WHO/HAI medicinal price methodology were used. The selection of survey areas and individual pharmacies was based on the WHO/HAI medicinal price methodology, as detailed in Supplementary File One (Supplementary File S1). A single medical officer of the health (MOH) area (the smallest health administrative area in Sri Lanka, which usually includes a population of 50,000–100,000) was considered a single survey area because it fulfills the requirements of a survey area according to the WHO/HAI price methodology (out of a total of 19 MOH areas in the district) [7]. The WHO/HAI outlines the selection of six geographical survey areas for pharmacy inclusion, including the area with the capital city and other areas, randomly based on accessibility within a day from the main city. Therefore, the MOH area, which includes the district's main urban center, and five other MOH areas were initially planned to be included in the study. Each survey area should comprise five private and five other sector pharmacies. Ratnapura City is part of the Ratnapura Municipal Council MOH area and served as one survey area. The remaining five MOH areas, accessible within a day from Ratnapura city, were randomly selected from all MOH areas in the Ratnapura District, as all were within a day's reach [7]. In our study, despite excluding government institutions, we rigorously adhered to the WHO/HAI medicinal pricing methodology to select private and "other" sector pharmacies. The first step in selecting private pharmacies from each selected MOH area involved identifying the government institutions as the private pharmacies closest to the respective government institution according to the WHO/HAI methodology [7]. The major government institutions in each MOH area were first selected, and an additional four government institutions were randomly chosen to equally represent each level of care/hospital category. In many MOH areas in Ratnapura, there were fewer than five government institutions per MOH area (3.8 per MOH area). Therefore, an adjacent MOH area was selected in addition to the randomly selected MOH area using convenience sampling. Thus, a single survey area consisted of two MOH areas (the final total of the MOH areas selected were 12 out of 19 available in the district). Finally, 30 private pharmacies were selected (5 per survey area). However, for the "other" sector, there were only two state retail pharmacies in the district, and both were included (Supplementary File S1).

An interviewer-administered checklist was used. This checklist was adapted from the guidelines provided by the WHO/HAI methodology for medicinal prices [7] (Supplementary File S2). Eleven essential oral paediatric anti-infective medicinal drugs were selected with the aid of the national list and the WHO model list of essential medicinal drugs by a committee of experts representing paediatric pharmacology [44,48]. According to the guidelines, the prices of OB and LPG were obtained for each surveyed medicine in each surveyed pharmacy. To obtain information on the prices of the selected medici-

nal drugs, a price checklist was developed based on the model checklist provided in the guidelines on WHO/HAI medicinal drug pricing methodology [7] (Supplementary File S2). The data collector, who was an undergraduate student in biological sciences, visited selected pharmacies. The data collector completed the WHO/HAI medicinal drug price checklist after providing informed consent. The MPR of the LPG and OB of each oral anti-infective medicine surveyed was calculated. This was calculated as per milliliter of oral liquid form/per tablet (median price ratio = median local price/international reference unit price). The international reference unit price was taken from the latest version of the MSH International Drug Price Indicator Guide. This is a widely used and routinely updated reference price indicator list available [6,45]. The acceptability of the prices of the anti-infectives was decided based on the MPR. The standard cutoff point of the MPR was 2.5. However, following expert opinion on paediatric pharmacology, 2 was selected as the cutoff. The 2.5 limit was introduced as a global value, considering high-income countries as well. With Sri Lanka being a low–middle-income country [49], we selected an MPR of 2, factoring the use of 2 as a cutoff by many medicinal affordability studies conducted in Asian countries [6,7,50–52]. The mean percentage difference in MPRs between OB and LPG was also calculated. These results were compared with those of a national survey on the prices of essential paediatric medicinal drugs carried out in 2009 [6] (mean percentage difference = MPR of OB—MPR of LPG/MPR of LPG × 100).

According to the WHO/HAI medicinal price methodology, the affordability of the treatment with the selected paediatric oral medicinal drugs relative to primary caregivers was approximated by the use of the average daily salary of the lowest-paid unskilled public worker and median local prices. The unit of measurement was the number of days of the daily salary of the lowest-paid unskilled public worker (LRK1213.67 or USD 6.06) needed for standard treatment with a first-line anti-infective medicine for a common acute childhood illness such as respiratory tract infection [14]. The standard dose recommended for a child in the 3–5 years age group was adopted for dose calculation. The treatment course for selected paediatric oral antibacterial, antiviral, antihelminthic, and antifungal medicines is described in Supplementary File Three (Supplementary File S3). As the salaries of public workers differ from those of the private sector, in addition to the lowest-paid unskilled public worker, the legally defined lowest-paid unskilled private worker daily salary (LRK 400.00 or USD 2.00, enacted in 2016) was also used for calculations [6,7,13,14,45].

## 5. Conclusions

With respect to essential oral paediatric anti-infective medicinal drugs for which maximum retail prices were controlled by the National Medicines Regulatory Authority, medicinal prices were within acceptable price levels and therefore safeguarded the financial access to primary caregivers of 3-year-old children in the Ratnapura District. However, for selected essential oral paediatric anti-infective medicinal drugs for which maximum retail prices were not controlled by the National Medicines Regulator Authority, the prices of the originator brand were above the acceptable level (MPR above 2). However, a recent economic crisis involving hyperinflation may negatively impact financial access if price control is not regularly updated based on the exchange rate and other associated costs. Therefore, we propose including a revision of pricings frequently, accounting for landing and other associated costs and the development of a transparent scientific criterion to subside "unaffordable" essential medicinal drugs.

**Supplementary Materials:** The following supporting information can be downloaded at https://www.mdpi.com/article/10.3390/pharma3010011/s1. Supplementary File S1: The initial definition of the survey area. Supplementary File S2: Paediatric Oral Anti-Infective Medicine Price Data Collection Form. Supplementary File S3: Dosage and Duration of Treatment for Selected Oral Paediatric Anti-infective Medicines for Affordability Analysis.

**Author Contributions:** Both M.K. and M.C.W. participated in the design of the study. M.K. coordinated the data collection and performed the statistical analysis. Both M.K. and M.C.W. interpreted

the data and drafted the first version of the manuscript. All authors have read and agreed to the published version of the manuscript.

**Funding:** This research received no external funding.

**Institutional Review Board Statement:** The study was conducted in accordance with the Declaration of Helsinki and approved by the Ethics approval and consent to participate. Ethics approval was granted by the Ethics Review Committee of the Faculty of Medicine, University of Colombo, Sri Lanka (protocol code EC-18-120 and 20 December 2018), for studies involving humans.

**Informed Consent Statement:** Informed consent was obtained from all subjects involved in the study. Written informed consent was obtained from the pharmacies to publish this paper.

**Data Availability Statement:** The data are available upon reasonable request. The raw data without personal identifiers are available from the corresponding author upon reasonable request.

**Acknowledgments:** We are grateful to the study participants, the data collectors for the study, and the Postgraduate Institute of Medicine, University of Colombo.

**Conflicts of Interest:** The authors declare that they have no competing interests.

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
