# Peer review of "Affordability of Paediatric Oral Anti-Infective Medicines in a Selected District, Sri Lanka"

_2813-0618, doi:10.3390/pharma3010011_

Round 1

Reviewer 1 Report

Comments and Suggestions for Authors

The manuscript presents a study to assess availability and affordability of essential paediatric formulations of anti-infectives in a single district in Sri Lanka. The authors have followed the standard WHO/HAI methodology to a good extent debarring the exclusion of public pharmacies though gave a valid reason for the same. The study is very well conducted and reported. The results have been elaboratively presented and discussed. No major comments from my side.

1.     Methods: Please provide reference for the sentence “The WHO 409 recommends the use of an international reference price index in the publication produced by Management Science for Health (MSH)”. 

Comments on the Quality of English Language

Minor editing required

Author Response

Reviewer

Comment

Authors’ Response

One

Minor editing of English language required

We have extensively language edited the whole document using professional service. The changes are highlighted in green.

Methods: Please provide reference for the sentence “The WHO 409 recommends the use of an international reference price index in the publication produced by Management Science for Health (MSH)”.

Thank you for highlighting!

We have added the reference (lines 414-416). It is highlighted in yellow.

The WHO recommends the use of an international reference price index in the publication produced by Management Science for Health (MSH)[36].

Reviewer 2 Report

Comments and Suggestions for Authors

Dear Authors,

This is an interesting and necessary research. I have just a couple of points:

1) Perhaps it would have been interesting to include other antiinfectives quite used in children, such as macrolides (azithromycin and clarithromycin). Also, sulfamethoxazole + trimethoprim. Perhaps explaining why these three antibiotics were not included would be useful. Especially because both macrolides belong to the Watch category of the WHO-AWaRe classification and sulfa+trim. belongs to the Access category.

2) In my opinion, one of the main findings is the lack of availability of nitrofurantoin and metronidazole. Both are Access antibiotics according to the WHO-AWaRe classification. Both are recommended in many infectious processes in children as a first-line treatment (according to the WHO AWaRe Antibiotic book). So, the clinical and public health consequences of their non-availability should be discussed and highlighted.

I would suggest adding the reference:

"Moja L, Zanichelli V, Mertz D, et al. WHO's essential medicines and AWaRe: recommendations on first- and second-choice antibiotics for empiric treatment of clinical infections. Clin Microbiol Infect. 2024 Feb 9:S1198-743X(24)00059-4. doi: 10.1016/j.cmi.2024.02.003. Epub ahead of print."

to refer to the WHO AWaRe classification of antibiotics and the WHO EML AWaRe handbook, where the most updated recommendations for adults and children are collected.

In my opinion, any study about affordability is incomplete if there is no reference to the findings' clinical and public health importance. My suggestion is to add this to the discussion and, perhaps, summarise it in a sentence in the Abstract.

Author Response

Reviewer

Comment

Authors’ Response

Two

1) Perhaps it would have been interesting to include other antiinfectives quite used in children, such as macrolides (azithromycin and clarithromycin). Also, sulfamethoxazole + trimethoprim. Perhaps explaining why these three antibiotics were not included would be useful. Especially because both macrolides belong to the Watch category of the WHO-AWaRe classification and sulfa+trim. belongs to the Access category.

Thank you for highlighting an important limitation. Yes expert committee and we considered a few medicines that included azithromycin, clarithromycin and  sulfamethoxazole + trimethoprim in addition to Mebendazole, and Cefuroxime. However, we omitted azithromycin, and clarithromycin because they were not included in the National List of  Essential Medicines Sri Lanka (4th edition). Nevertheless, we discussed that the omissions could act as a limitation. Sulfamethoxazole + trimethoprim was omitted because it was not widely used in Sri Lanka compared to other anti-infective medicine drugs included in the survey (according to the pediatricians who participated in the expert committee). Therefore, it was omitted from the final list.

However, we acknowledge the limitations due to these omissions. And we have added your concern as a limitation in the discussion (lines 394-400).

Further, we limited ourselves to 11 key essential paediatric anti-infective medicines and excluded WHO recommended essential medicines such as azithromycin, clarithromycin, and sulfamethoxazole + trimethoprim [36]. This decision was based on the “National List of Essential Medicines Sri Lanka (4th edition)” [37] and the opinion of the expert committee on the selection of anti-infective medicines for the study.

2) In my opinion, one of the main findings is the lack of availability of nitrofurantoin and metronidazole. Both are Access antibiotics according to the WHO-AWaRe classification. Both are recommended in many infectious processes in children as a first-line treatment (according to the WHO AWaRe Antibiotic book). So, the clinical and public health consequences of their non-availability should be discussed and highlighted.

I would suggest adding the reference:

"Moja L, Zanichelli V, Mertz D, et al. WHO's essential medicines and AWaRe: recommendations on first- and second-choice antibiotics for empiric treatment of clinical infections. Clin Microbiol Infect. 2024 Feb 9:S1198-743X(24)00059-4. doi: 10.1016/j.cmi.2024.02.003. Epub ahead of print."

to refer to the WHO AWaRe classification of antibiotics and the WHO EML AWaRe handbook, where the most updated recommendations for adults and children are collected.

In my opinion, any study about affordability is incomplete if there is no reference to the findings' clinical and public health importance. My suggestion is to add this to the discussion and, perhaps, summarise it in a sentence in the Abstract.

Thank you for highlighting this important issue. We have included a paragraph on the availability of these two medications referring the WHO essential medicines list (lines 181-194 & 394-398).

Four key essential medicines demonstrated poor availability of less than 50% for generics in the private sector (Table 1). This could potentially impact the access to affordable and essential medicines for children for common infections and could cause significant harm to the paediatric population in the district. Further, the unavailability of nitrofurantoin in liquid form (either generic or OB), reported in both private and Osu-Sala pharmacies, could pose a significant risk to the health of children in the district (Table 1). Nitrofurantoin is one of the first lines of treatment for uncomplicated lower urinary tract infections (UTI) in children as well as for prophylaxis from UTI [16]. Similarly, the availability of Metronidazole in liquid form in the district is less than 50% (Table 1). Metronidazole is being used as one of the first lines of treatment for mild-to-moderate Clostridium difficile-associated disease in children which is a common cause for gastroenteritis in children [17]. Both metronidazole and nitrofurantoin are listed as essential medicines by the WHO underlining the importance of universal availability for the treatment of children and nonavailability is a public health issue that requires urgent intervention [18].

Further, we limited ourselves to 11 key essential paediatric anti-infective medicines and excluded WHO recommended essential medicines such as azithromycin, clarithromycin, and sulfamethoxazole + trimethoprim [18].

Reviewer 3 Report

Comments and Suggestions for Authors

This study utilized the private and other sector retail price components of the WHO/HAI medicinal price methodology to assess the affordability of oral pediatric anti-infective medications in the Ratnapura District of Sri Lanka. The study demonstrates commendable methodological rigor by employing the established WHO/HAI medicinal price methodology, ensuring robustness in assessing medication affordability. The research provides valuable insights by focusing on the Ratnapura District, enhancing the understanding of medication accessibility within this specific geographical context.

In the article, the following issues should be addressed:

1.       L14 The fact that only 50% of private pharmacies offer all 11 medicinal drugs in generic form may indeed contribute to a reduction in the accessibility of medications. This limitation arises from the potential decrease in options for patients who specifically seek generic versions of drugs due to cost considerations or other factors.

2.       L43-44 please explain why select the lowest paid unskilled public worker

3.       L56,120,127  “Health Survey or DHS (2016)”, “the pirce of Pyrantel” and “The daily salary for the private sector” need the literature source

4.       L159 Figure1 Adding a standard line for MPR (Maximum Price Ratio) in Figure 1 would indeed enhance reader comprehension.

5.       What were some of the disparities in availability of OB and LPG in private pharmacies?

6.       L263 The MPRs of all these three medicinal drugs are above 2, Based on the findings, should consider implementing measures to improve the affordability and accessibility of essential pediatric medications

7.       L477 The standard of MPR was 2.5, this study slect 2 in Sir Lankan situation. Please provide more explanation.

8.       L373-376 The study was conducted in the wake of COVID-19 in 2021 and the current economic crisis in Sri Lanka Whether there is a serious effect on the results of the study

9.       Authors should add some discussion about the value of this study to other areas or countries in the world.

Comments on the Quality of English Language

Extensive editing
